# Volatile Organic Compounds in Underground Shopping Districts in Korea

**DOI:** 10.3390/ijerph18115508

**Published:** 2021-05-21

**Authors:** Soo Ran Won, Young Sung Ghim, Jeonghoon Kim, Jungmin Ryu, In-Keun Shim, Jongchun Lee

**Affiliations:** 1Indoor Environment and Noise Research Division, National Institute of Environmental Research, Incheon 22689, Korea; wsr1984@korea.kr (S.R.W.); kimxls88@gmail.com (J.K.); mynamin@korea.kr (J.R.); inkeun77@korea.kr (I.-K.S.); roundvoid@korea.kr (J.L.); 2Department of Environmental Science, Hankuk University of Foreign Studies, Yongin 17035, Korea

**Keywords:** indoor air quality, volatile organic compound, principal component factor analysis, source apportionment, carcinogenic risk

## Abstract

Underground shopping districts (USDs) are susceptible to severe indoor air pollution, which can adversely impact human health. We measured 24 volatile organic compounds (VOCs) in 13 USDs throughout South Korea from July to October 2017, and the human risk of inhaling hazardous substances was evaluated. The sum of the concentrations of the 24 VOCs was much higher inside the USDs than in the open air. Based on factor analysis, six indoor air pollution sources were identified. Despite the expectation of a partial outdoor effect, the impacts of the indoor emissions were significant, resulting in an indoor/outdoor (I/O) ratio of 5.9 and indicating elevated indoor air pollution. However, the effects of indoor emissions decreased, and the contributions of the pollution sources reduced when the USD entrances were open and the stores were closed. Although benzene, formaldehyde, and acetaldehyde exhibited lower concentrations compared to previous studies, they still posed health risks in both indoor and outdoor settings. Particularly, while the indoor excess cancer risk (ECR) of formaldehyde was ~10 times higher than its outdoor ECR, benzene had a low I/O ratio (1.1) and a similar ECR value. Therefore, indoor VOC concentrations could be reduced by managing inputs of open air into USDs.

## 1. Introduction

Hazardous materials may cause severe indoor air pollution in underground shopping districts (USDs). This is because USDs have many different types of stores due to the characteristics of the facilities, and because ventilation facilities are often inadequate. Various types of goods, furniture (e.g., chairs and tables), construction materials (e.g., paint), and household electrical appliances (e.g., computers) emit a variety of volatile organic compounds (VOCs). These compounds are present in indoor air and can adversely affect indoor air quality [1,2,3]. To address this, the Korean Ministry of Environment (MOE) enacted the “Underground Living Space Air Quality Management Law” in 1996, which aims to manage the air quality in the living spaces of underground facilities (e.g., USDs and subways) to protect residents from exposure to hazardous materials. The law is known as the “Indoor Air Quality Control Act”, in which USDs are classified as “public-use facilities”, and hazardous materials such as formaldehyde, total volatile organic compounds (TVOC), particulate matter with an aerodynamic diameter < 10 µm (PM_10_) and <2.5 µm (PM_2.5_), and CO_2_ are designated as target substances with the objective of managing indoor air quality. Public-use facilities are defined as facilities that are utilized by unspecified groups of people [4]. In South Korea, public-use facilities are classified into 24 types, and the concentrations of hazardous substances are classified and managed based on the characteristics of each facility type.

Benzene, formaldehyde, and acetaldehyde are classified as carcinogenic substances belonging to Groups A, B1, and B2, respectively, in the United States Environmental Protection Agency (EPA) Integrated Risk Information System (IRIS); they have been identified as substances that pose potential risks to the human body [5]. Formaldehyde and acetaldehyde are emitted by construction materials, fibers, and automobile exhaust fumes, and exposure may result in eye, nose, and throat irritation, nausea, and cytotoxic irritation of the respiratory tract and eyes [6,7]. VOCs (such as benzene and toluene) are generally emitted by construction materials, laundry solvents, and insecticides and may lead to breathing difficulties, sleepiness, headache, and building syndrome [8,9,10,11,12]. Moreover, VOCs in the atmosphere are the precursors of secondary particles and gases [13]. 

As in other countries, USDs for pedestrian traffic have been constructed in Korea to facilitate the passage of citizens while constructing underground stations and large buildings. USDs enable efficient use of space in large-scale urban centers and are also equipped with commercial facilities. However, in Korea, where south and north are divided, USDs were also used for evacuation purposes during the war. Previous studies on formaldehyde and VOCs have focused on measuring the concentrations of hazardous materials in indoor air produced by indoor pollution sources, while others have focused on outdoor pollution sources, indoor air quality in living spaces, risk assessments of exposure to hazardous materials, and pollution source estimation [8,14,15,16,17,18,19,20,21]. Research on indoor air quality in USDs has been conducted widely in East Asia, particularly in China and Hong Kong. These studies have focused on USDs and large department stores where various goods are sold, which are similar to USDs [11,22,23,24,25,26]. Although some studies have focused on measuring hazardous materials in indoor air in various public-use facilities (including USDs), few have focused specifically on USDs [2,3,27,28]. In this study, the indoor and outdoor concentrations of VOCs in South Korean USDs and their associated sources were investigated, and the risks of inhaling carcinogenic substances (such as benzene and formaldehyde) were assessed.

## 2. Methods

### 2.1. Study Sites

We investigated 13 USDs in 7 regions of South Korea (Figure 1). Measurements were conducted from July to October 2017 at 2–3 indoor locations in each USD, along with 1 location in the open air circumjacent to the USD (depending on the area). The USDs were constructed between 1978 and 2009, and their areas ranged from 1,030 to 19,756 m^2^. The number of shops in each USD (excluding empty rooms) ranged from 16 to 319 and included clothing stores (56%), followed by shops selling shoes or bags (8%), electronics stores selling mobile phones (6%), food stores such as restaurants or cafes (3%), cosmetics shops (2%), and nail shops (1%). Other unclassified stores such as convenience stores, institutions, and flower shops accounted for 24% of the stores. We divided the USDs into “open,” “semi-open,” and “closed” types, depending on the type of the entrance to the outdoors. The open type had no door at the entrance of the USD, and air could be exchanged freely through the entrance. The semi-open type had one door at the entrance that partially blocked the air flow. The closed type had a double door or an air curtain to separate the indoor and outdoor air. We also divided USDs into “open,” “mixed,” and “closed,” depending on the store entrance opening up to the passageway. For closed-type facilities, stores were separated from the passageway, whereas for open-type facilities, stores were connected directly to a passageway and had no entrance. In mixed-type facilities, stores with and without entrances coexisted.

Because USDs were dominated by open-type stores, the VOC characteristics of each store type were difficult to determine, even if measurements were made at each store. Therefore, for separate stores (i.e., (1) clothing stores; (2) fashion accessory (leather goods) stores; (3) electronics stores; (4) restaurants; (5) cafes; (6) cosmetics shops; (7) nail shops) that could exhibit distinct VOC characteristics from the USD, indoor VOC concentrations were measured for 7 days at one indoor location. Although all seven store types were located adjacent to the road at ground level, the entrances were kept closed at all times. The areas of these stores ranged from 24.5 to 94.3 m^2^, and we assumed that remodeling did not have a significant impact, as it was completed over 6 months prior to the measurement period. An air conditioner was operated in four of the seven stores, and cleaning was carried out 1–2 times per day (vacuum and wet cleaning). 

### 2.2. Sampling and Analyses

We determined the number of sampling points according to the standard methods for indoor air quality specified by the National Institute of Environmental Research, Korea [29]. We sampled two points for areas <10,000 m^2^, three for areas between 10,000 and 20,000 m^2^, and four for areas >20,000 m^2^. We sampled outdoor air at one point within 10 m distance the entrance of each USD. The sampling period spanned from July to October 2017, and sampling was performed on three consecutive days for each USD.

Although we measured 58 VOCs, we analyzed 24 VOCs. We excluded 34 VOCs for which the concentrations were sometimes below the detection limits. The 24 VOCs analyzed in this study were aromatics (benzene, toluene, ethylbenzene, xylenes, and styrene), aliphatics (n-heptane, n-octane, n-nonane, n-decane, n-undecane, n-dodecane, n-tridecane, n-tetradecane, and n-pentadecane), terpenes (α-pinene and d-limonene), oxygenated VOCs (nonanal, n-butanol, formaldehyde, acetaldehyde, acetone, propionaldehyde, butyraldehyde, and benzaldehyde), and total volatile organic compounds (TVOC). Of these, benzene, toluene, ethylbenzene, xylenes, styrene, n-heptane, n-octane, n-nonane, n-decane, n-undecane, n-dodecane, n-tridecane, n-tetradecane, n-pentadecane, nonanal, n-butanol, α-pinene, and d-limonene were measured twice in 30 min intervals at a 100 mL/min flow rate using an MP-∑30 (Sibata, Japan) pump fitted with a sorbent tube (Supelco, Bellefonte, PA, USA) filled with 200 mg of Tenax-TA. In the sorbent tube, a tube conditioner (ATC 1200, ACEN, Suwon, Korea) was used for the desorption of high-purity nitrogen gas (99.999%) at a 100 mL/min flow rate at 320 °C for 180 min. After sample collection, the sorbent tube was closed with a ¼ inch thread-type stopper filled with polytetrafluoroethylene (PTFE) ferrules, sealed with Parafilm, and was refrigerated until analysis. To account for any contamination that may have occurred during transport, field blanks were pre-treated, transferred, stored, and analyzed under the same conditions as the samples and were used for correcting the measured values. 

The VOCs collected using Tenax-TA were analyzed using thermal desorption‒gas chromatography mass spectrometry (TD‒GCMS) with a GC-2000 Plus (Shimadzu, Kyoto, Japan). The analytical conditions are listed in Table 1. Five standard solutions (i.e., 20, 50, 100, 200, and 500 μg/mL) were vaporized in the Tenax-TA tube at 200 °C with 100 μg/mL of an indoor air standard mixture (Supelco, Bellefonte, PA, USA) and mixed at 100 mL/min for 5 min. The 20 and 50 μg/mL solutions were diluted in methanol, and 1 μL was injected into the sorbent tube; the 100, 200, and 500 μg/mL standards were undiluted and volumes of 1, 2, and 5 μL were injected, respectively. The coefficient of determination (R^2^) of the calibration curve exceeded 0.99, and the method detection limits (MDL) ranged from 0.90 ng (n-heptane) to 5.14 ng (nonanal). TVOC concentrations were calculated by applying the response factor (RF) of toluene to the sum of the total peak area eluted from n-hexane to n-hexadecane [30].

Formaldehyde, acetone, acetaldehyde, butyraldehyde, benzaldehyde, and propionaldehyde were collected twice for 30 min at a flow rate of 500 mL/min using an MP-∑100 (Sibata, Saitama, Japan) pump and cartridge (Supelco, USA) filled with silica gel coated with 2,4-dinitrophenylhydrazine (DNPH). By installing an ozone scrubber (Waters, Milford, MA, USA) filled with high-purity potassium iodide (KI) at the cartridge intake, the interference of ozone (which reacts with DNPH derivatives) was reduced. After sample collection, the cartridge was closed with the stopper previously enclosed with the cartridge, sealed with Parafilm, stored in an aluminum zipper bag to prevent contact with light, and refrigerated at 4 °C until analysis. Using cartridges with the same lot number as those used for sample collection, field blanks were transferred, stored, pre-treated, and analyzed with the sample cartridges and were used for correction after analysis. 

For each cartridge, 5 mL of 2,4-DNPH derivatized aldehyde was extracted using acetonitrile into a volumetric flask with a vacuum manifold (Supelco, Bellefonte, PA, USA), i.e., solid-phase extraction. The extracted solutions were characterized using high-performance liquid chromatography (HPLC; Ultimate 3000, Thermo Fisher Scientific, Waltham, MA, USA). The column used for analysis was an Allure^TM^ C18 (150 × 4.6 × 5 μm, Restek, Centre County, PA, USA), and acetonitrile and water at a ratio of 7:3 (*v*/*v*) were added at a rate of 1.0 mL/min. The column oven temperature was set to 30 °C, and detection was conducted at a UV wavelength of 360 nm. For the standard solutions, a TO11/IP-6A aldehyde/ketone-DNPH Mix 15 μg/mL solution (Supelco, Bellefonte, PA, USA) was diluted with acetonitrile and analyzed in five steps (0.03, 0.06, 0.12, 0.30, and 0.60 μg/mL). The R^2^ of the analytical calibration curve exceeded 0.99, and the MDL at different analytical conditions ranged from 0.0004 μg/mL (formaldehyde) to 0.0017 μg/mL (benzaldehyde). Note that the MDL of carbonyl compounds is given in μg/mL, whereas that of other VOCs is given in ng; this is because it depends on measurement and analysis methods, which differ by compound.

### 2.3. Data Analyses

To estimate the VOC sources, SPSS for Windows 20.0 (IBM, Armonk, NY, USA) was used to conduct factor analysis with varimax rotation extraction using descriptive statistical analysis, correlation analysis, and principal component analysis of the VOCs. Furthermore, we examined the variations in group means with the USD environmental factors such as the region and entrance, store shape, and the number of stores, to estimate the contribution of each VOC source. 

For benzene, formaldehyde, and acetaldehyde (i.e., high-risk carcinogens), the excess cancer risk (ECR) was assessed using the following equations [31]:EC = (CA × ET × EF × ED)/AT
ECR = EC × IUR
where EC denotes the exposure concentrations via inhalation pathway, and CA represents contaminant concentration in air (95th percentile concentration in this study). ET, EF, and ED denote exposure time, exposure frequency, and exposure duration, respectively. For a typical USD store owner scenario, ET, EF, and ED are assumed to be 12 h/day (open 10 am to 10 pm); 350 days per year (two weeks off for holidays); 15 years (conservatively assumed two extensions of original 5-year lease), respectively. AT represents averaging time in lifetime (82 years by life expectancy of Korean). The IUR are obtained from toxicity value of EPA’s IRIS.

## 3. Results and Discussion

### 3.1. VOC Concentrations of Indoor and Outdoor Air

The VOC concentrations measured at 30 locations in 13 USDs throughout South Korea are presented in Table 2. The formaldehyde concentrations in USDs sampled in this study were below the threshold (100 μg/m^3^) for public-use facilities regulated by the Indoor Air Quality Control Act of the Ministry of Environment; however, two of the 30 locations had formaldehyde concentrations that exceeded the threshold. Although the TVOC concentrations were below the threshold (500 μg/m^3^) in 11 of the 13 USDs, two USDs had concentrations above the threshold. There were stores nearby selling fashion accessories, clothing, and other general goods in two USDs where the concentration of formaldehyde and TVOC exceeded the thresholds. In addition, chemical scent such as adhesives were detected during measurement. It was measured in the passage of the USDs, but since 70–80% of the stores are the open type, it was estimated that the effect of the stores had affected the passage.

In the previous studies of hazardous materials in shopping malls and department stores (i.e., facilities similar to USDs), formaldehyde concentrations were similar to those obtained in this study [22,23]. The concentrations of benzene, toluene, ethylbenzene, xylenes, styrene, and TVOC obtained in this study were lower than those in China and Hong Kong; however, they were similar to those in Europe [11,24,32,33]. According to Lee et al. [27], the concentrations of formaldehyde and TVOC in the indoor air of South Korean USDs in 2005–2006 were 35.8 and 2267 μg⁄m^3^, respectively, which are higher than those obtained in the present study. During that time, indoor air quality did not have a significant impact on citizens; thus, the TVOC concentration greatly exceeded the threshold. However, the amount of time spent indoors has since increased, and the effects and importance of indoor air quality have consequently increased; therefore, the relevant regulations have been strengthened. The standards for outdoor and indoor air quality have also become more stringent. However, only the benzene concentration of VOCs is regulated as hazardous substances in outdoor air, while other VOCs are regulated as ozone.

In open air, the concentrations of benzene, toluene, ethylbenzene, xylenes, and styrene measured in Seoul in 1996 [34] were not significantly different from the results obtained in the present study. According to Na et al. [34], the major emission source in downtown Seoul was assumed to be automobile exhaust fumes. Owing to their high photochemical reactivity, such fumes are expected to contribute to the formation of secondary aerosols such as smog. In the present study, because USDs were located at city centers and the surrounding traffic was considerable, the effects of exhaust fumes were likely to be significant. 

The indoor/outdoor (I/O) ratios of temperature, formaldehyde, and VOCs (except relative humidity) all exceeded 1. Compared to open air, indoor temperatures had relatively small deviations, and the relative humidity was lower; thus, suitable temperature and humidity conditions were maintained by air conditioning equipment in the USDs. Except for benzene and propionaldehyde, the I/O ratios of the VOCs exceeded 2, indicating that the effects of indoor pollution sources were more significant than those of open air sources. Benzene and propionaldehyde were highly correlated with open air, which suggested that they originated from open air emissions. Benzene, xylenes, ethylbenzene, and toluene were reported to be derived from automobile exhaust fumes [14,35]. In this study, xylenes was highly correlated with both indoor and outdoor sources; however, it had a high I/O ratio, suggesting that the effects of indoor pollution sources were more significant. Propionaldehyde was reported to be weakly correlated with open air [36]. Propionaldehyde has been reported as being derived from petrochemical industry emissions [20]. Santarsiero and Fuselli [19] claimed that propionaldehyde is produced by indoor kitchen-related activities or is introduced to indoor areas due to open air arboreal plants or artificial effects. In the open air, artificial effects are based on the ratio of acetaldehyde to propionaldehyde (C_2_/C_3_). In this study, the C_2_/C_3_ was approximately 2.65, which is similar to the results of Santarsiero and Fuselli [19] and suggests that propionaldehyde was introduced to indoor areas due to artificial effects. 

### 3.2. VOC Concentrations in Selected Stores

Based on the I/O ratios of VOCs in the USDs, the VOC contents were dominated by the types of stores in the USDs, the products sold at these stores, and the construction materials used in the USDs. Accordingly, the formaldehyde and VOC concentrations were measured in seven representative stores that reflected the types of stores in USDs. Formaldehyde concentrations were the highest in clothing stores, followed by nail shops, cafes, and electronics stores (Figure 2). TVOC concentrations were the highest in fashion shops, followed by nail shops, cafes, electronics stores, and clothing stores. As formaldehyde is produced by fabrics and adhesives in clothing shops, the observed high concentrations in these stores were not surprising. 

Formaldehyde concentrations were high in furniture shops that contained lumber products [23]. In the study from Tao et al. [24], formaldehyde concentrations decreased in the order of fashion accessory stores > restaurants > cosmetics shops, and TVOC concentrations decreased in the order of restaurants > cosmetics shops > fashion accessory stores. It was reported that VOC concentrations were high in fast food and fashion accessory stores, where gas stoves are used or frequent cleaning occurs [11]. Formaldehyde concentrations were below the Korean threshold (100 μg/m^3^) in the aisles of USDs [3]; however, they were above this threshold in various types of stores, such as clothing and shoe shops (Figure 2). Although the TVOC concentrations were also below the threshold in aisles, they were two-fold higher than the threshold in shoe stores, suggesting contamination due to goods for sale. Sim et al. [2] determined that the use of adhesives in the goods in stores selling shoes and bags, along with chemicals at nail shops and cosmetics shops, were sources of VOCs. 

### 3.3. Source Apportionment of Indoor VOCs

Factor analysis was conducted using principal component analysis of the 24 types of indoor VOCs analyzed in this study. Six groups had eigenvalues > 1 and accounted for 85.6% of the cumulative variation (Table 3). Prior to specifying the pollution sources for each factor, the characteristics of each factor were investigated using the correlation coefficients between indoor and outdoor concentrations and the I/O ratios presented in Table 4. We assumed that each factor was composed of marker species and that the concentration of each factor was the sum of the concentrations of the marker species. The open air concentration was calculated as the sum of all 24 components, and the I/O ratio was calculated as the sum of all 24 components (both indoors and outdoors). 

Factor 1 had high loadings of acetone, butyraldehyde, n-pentadecane, benzaldehyde, toluene, d-limonene, n-butanol, formaldehyde, styrene, and acetaldehyde and accounted for 28% of the total variation. Because materials with high I/O ratios, such as acetone, formaldehyde and acetaldehyde (oxygenated VOCs), styrene (aromatics), and d-limonene (terpenes), were included (Table 2), factor 1 was significantly affected by indoor effects, as shown in Table 4. In addition to the observed high formaldehyde, acetone, and styrene emissions in all store types (Table 3), formaldehyde was emitted by building materials, smoking, and household products [1,37,38]. Acetone and acetaldehyde were emitted by combustion, household products, and smoking [1,37,38,39,40]. This was because, although smoking was prohibited in public-use facilities (including USDs), it was not strictly enforced. Toluene, which is a major component of automobile emissions, was also emitted by all store types (Table 3). In other words, factor 1 sources (including building materials, combustion, and automobile emissions) were universal in both indoor and outdoor areas; however, the proportion of indoor emissions was higher. 

Factor 2 had high loadings of n-tetradecane, benzene, n-undecane, n-dodecane, and n-tridecane and accounted for 15% of the total variation. Benzene is emitted by almost all processes involving VOCs, including volatilization of gasoline and automobile emissions [1,39,41]. Decane is widely utilized in diesel vehicles, floor and wall coverings, industrial coatings, and in the pulp and paper industry [1,39,40]. In other words, benzene is mainly involved in fuel utilization, whereas decane is used in exterior wall decorations such as coatings and coverings. Both benzene and decane are universal. Although factors 1 and 2 were both universal, their correlation was low (0.27), and the I/O ratio exceeded 1, indicating a large contribution from indoor emissions. This phenomenon can be explained by the fact that factor 1 depends on the characteristics of the store, whereas factor 2 is attributed to the universal characteristics of USDs, such as fuel utilization and exterior wall decorations. For these reasons, although both factors 1 and 2 exhibited high indoor emissions contributions, factor 2 had the lowest correlation with outdoor VOCs (0.012). 

Factor 3 contained high loadings of aliphatics (n-nonane, n-heptane, and n-octane) and accounted for 13% of the total variation. Nonane, heptane, and octane are utilized in floor and wall coverings and are emitted from the volatilization of gasoline and diesel, which is similar to factor 2 [1,14]. Accordingly, the correlation between factors 2 and 3 was 0.31, which was the second highest after the correlation between factors 1 and 5. However, the correlation of factor 3 with outdoor VOCs was also high (0.184), and the I/O ratio was low (1.86). In other words, emissions of factors 2 and 3 were similar; however, the contributions of indoor and outdoor emissions were higher for factors 2 and 3, respectively. 

Factor 4 had high loadings of α-pinene, nonanal, and propionaldehyde and accounted for 11% of the total variation. Although pinene has the potential for external factors, such as biogenic or landfill emissions [1,39], the I/O ratio of factor 4 was higher than that of factor 1, and its correlations with the other factors (including factor 1) were high, suggesting a higher contribution of indoor emissions. Terpenes such as α-pinene and d-limonene are used for cleaning products and air fresheners of oil or scent formulations, due to the favorable odor and solvent properties of terpenoids [42]. The pollution sources of oxygenated VOCs such as nonanal and propionaldehyde are not well known, except that their indoor emissions are significant. Nonanal are emitted not only by photochemical reactions of alkenes, but also by the combustion of plants [43]. Propionaldehyde also forms through the incomplete combustion of fossil fuels (including by automobiles) and by reacting with ozone indoors [44]. The high correlation between factor 4 and outdoor VOCs may have been due to external factors such as photochemical reactions and automobile emissions. Since USDs are generally located in densely populated city centers, it is necessary to investigate the impact of sources such as industrial or traffic emission plans in the future.

Factor 5 had a high n-decane loading and accounted for 11% of the total variation. N-decane was also included in factor 2 and is emitted by solvent use, volatilization of gasoline, and surface coatings [39,45,46]. The correlations between factor 5 and the other factors were not particularly high or low, its correlation with outdoor VOCs was not high, and the I/O ratio was relatively high, indicating a large contribution from indoor emissions. 

Factor 6 had high loadings of ethylbenzene and *m,p,o*-xylenes and accounted for 8% of the total variation. Although the I/O ratio exceeded 1, the correlation with outdoor VOCs was also high, likely due to the high emission contributions from fossil fuels [39,41]. However, the contribution from indoor emissions remained high, which may be attributed to emissions during painting and printing processes [39].

### 3.4. Contributions of VOC Sources by USD Environmental Factors

Table 5 shows the VOC concentrations and the contributions of each source to the different USD environmental factors. Because the total I/O ratio of the 24 VOCs was 5.88, as shown in Table 2 (note: Table 5 shows the overall value), the majority of the I/O ratios exceeded 1 regardless of the environment. The I/O ratio (6.2) for USDs with semi-open entrances (and consequently, limited ventilation) was higher than that for USDs with open entrances (Table 5(c)). Overall, the contribution of store emissions was the highest at 78%, that of indoor fuel was 6%, and that of combustion was 5%, accounting for approximately 90% of the total (Table 5(a)). This shows that store emissions and indoor fuel that included hazardous substances contributed significantly to VOC emissions. The contribution of store emissions composed of substances with high carcinogenic risks was high in densely populated areas such as Daejeon, Gyeonggi, and Busan, while the contribution of indoor fuel was the highest in Suncheon, where the total VOC concentrations were low (Table 5(b)). The high contribution of store emissions in USDs in densely populated areas was likely due to the larger number of stores in the Daejeon, Gyeonggi, and Busan USDs (101, 450, and 427, respectively). However, the contribution of store emissions in the Daejeon USD was high, even though it had fewer stores than the Gyeonggi and Busan USDs. This is likely due to the high fraction of open-type stores in the Daejeon USD (i.e., 86 of 101 stores were the open type). The contributions of store emissions decreased to 83, 74, and 67% when the store types were open, mixed, and closed, respectively (Table 5(d)).

Increases in the number of stores resulted in increases in the total concentrations of the 24 VOCs; however, these increases were not due to constant changes in pollution source contributions. By increasing the number of stores, the store types diversified, and consequently, the pollutants generated by these stores also diversified (Table 5(e)). The USD entrance and store shape had opposite effects, since the effect of outdoor emissions was significant when the entrance was open; however, the effect of emissions inside the store was reduced when the store shape was closed. However, when the entrance was open, the effect of outdoor emissions (such as fossil fuel combustion) along with indoor emissions (such as store emissions and indoor fuel use) decreased because the I/O ratio of the outdoor emissions also exceeded 1 and the contribution of indoor emissions was significant (Table 4 and Table 5(c)). Reductions in all emission contributions occurred when the store was of the closed type due to the decreased effect of indoor emissions, as mentioned above (Table 5(d)).

### 3.5. Excess Cancer Risk

In the EPA IRIS, carcinogens are divided into Groups A (human carcinogen), B1 (probable carcinogen, limited human evidence), B2 (probable carcinogen, sufficient evidence in animals), C (possible human carcinogen), D (not classifiable), and E (evidence of non-carcinogenicity) [5]. According to the International Agency for Research on Cancer, carcinogens are divided into Groups 1 (carcinogenic), 2A (probably carcinogenic), 2B (possibly carcinogenic), 3 (not classifiable), and 4 (probably not carcinogenic) [47]. Benzene is classified as a substance that is carcinogenic to humans (Group A) for all exposure paths, according to the 1986 Guidelines for Evaluating Carcinogenic Risk, and leads to leukemia when exposure occurs via inhalation [48]. Formaldehyde (Group B1) and acetaldehyde (Group B2) are classified as substances that are potentially carcinogenic to humans through inhalation exposure and lead to squamous cell carcinoma and adenocarcinoma [49,50]. In this study, the ECR was calculated based on the inhalation toxicity value provided by EPA IRIS and the indoor and outdoor 95th percentile concentrations of pollutants (Table 6). According to IRIS, one toxicity value is assigned to formaldehyde and acetaldehyde, whereas two values are assigned to benzene. Therefore, toxicity values of benzene were expressed as a range [51].

The EPA sets goals for ECR of 10^−6^ and 10^−4^ based on the maximum concentration near the pollutant source (1 and 100 persons per million, respectively) [52]. In this study, benzene, formaldehyde, and acetaldehyde presented ECRs > 10^−6^ indoors; they were present at levels at which the carcinogenic risk could not be regarded as negligible. In particular, the 95th percentile concentration of formaldehyde was approximately equivalent to the threshold stated in the Indoor Air Quality Control Act. However, the ECR indicates a close level that requires immediate action. In order to manage the formaldehyde concentration, it is necessary to prepare a management plan for periodic ventilation and operation of air conditioners in the USDs. Weng et al. [26] found that the 95th percentile risk of formaldehyde in a shopping mall was 3.0 × 10^−3^, which was higher than the value obtained in the present study.

The indoor and outdoor ECRs of benzene and acetaldehyde were not as high as those of formaldehyde; however, these pollutants need to be managed through continuous monitoring. In the USDs, benzene did not have a significant pollution source indoors and showed a high correlation with the open air. Hence, controlling benzene concentrations in the atmosphere is expected to be sufficient to reduce its risk indoors.

## 4. Conclusions

In this study, VOC sources were estimated using indoor and outdoor concentrations of 24 VOCs for the purpose of managing indoor air quality in 13 USDs in South Korea. In general, formaldehyde and TVOC concentrations were below their respective thresholds and the concentrations reported in previous studies. This suggests that the indoor air quality in the studied USDs is generally well managed. Although the majority of VOCs had high I/O ratios due to the effect of indoor pollution sources, benzene and propionaldehyde were significantly affected by artificial impacts of open air sources such as automobiles and coal power generation circumjacent to the USDs.

Based on factor analysis, the sources of indoor VOCs in USDs were of six types, including (1) emissions within the store, (2) indoor fuel use, floor and wall coverings, (3) outdoor fuel use, floor and wall coverings, (4) fossil fuel combustion and cleaning products, (5) solvent use and surface coatings, and (6) vehicle emissions and painting/printing products. The contributions of store emissions and indoor fuel (including materials with high carcinogenic risks) were high (78 and 6%, respectively). Because the effect of indoor emission was significant, the effect of open air was insufficient to significantly impact indoor air quality.

Highly carcinogenic benzene, formaldehyde, and acetaldehyde appeared to be present at harmful levels both inside and outside of USDs. The ECR of the 95th percentile concentration of benzene, formaldehyde, and acetaldehyde indicates that these substances should be under continuous observation. Benzene had a high correlation with the open air; therefore, its indoor risk would be reduced if the benzene concentrations in the open air were controlled.

This study was based on data collected in the aisles of USDs and identified the VOC concentrations in these facilities. However, measurements were not conducted inside the stores, which limits the interpretations of the data. In future studies, additional measurements in stores selling various products should be conducted in USDs to identify the effects in different types of stores.

## Figures and Tables

**Figure 1 ijerph-18-05508-f001:**
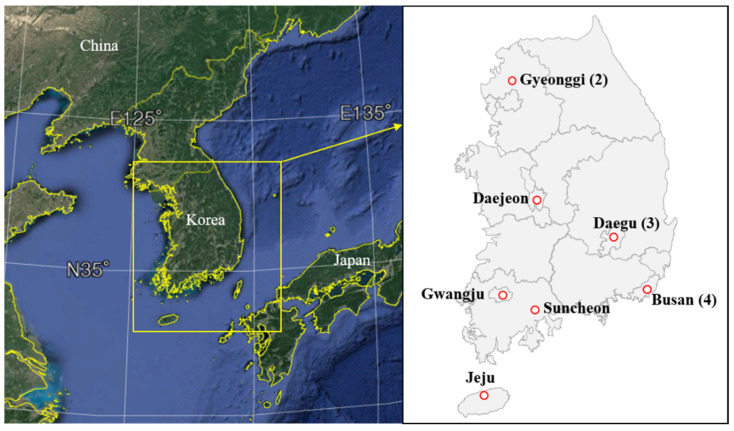
Distribution of 13 underground shopping districts (USDs) in South Korea selected for this study. Parentheses indicate the number of USDs analyzed in each area. The map on the left was generated using Google Earth.

**Figure 2 ijerph-18-05508-f002:**
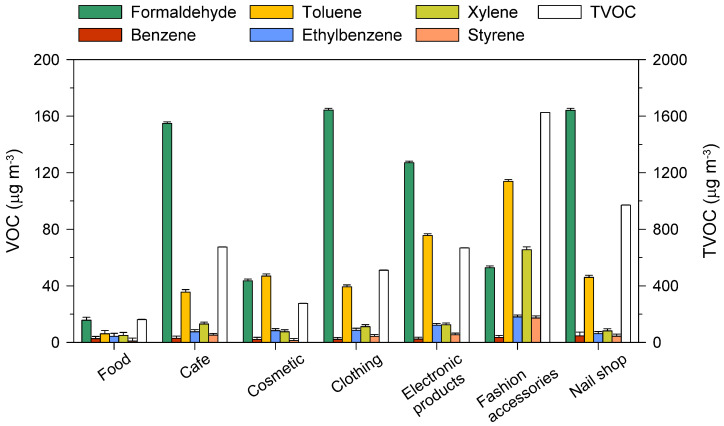
Concentrations (μg/m^3^) of frequently detected volatile organic compounds (VOCs) and TVOC in selected stores. Bar heights represent geometric mean concentrations; error bars in the positive direction show geometric standard deviations.

**Table 1 ijerph-18-05508-t001:** Thermal desorption gas chromatography/mass spectrometry (TD‒GC/MS) analytical conditions.

Parameter		Conditions
TD	Desorption time and flow	15 min, 60 mL/min
	Desorption temperature	280 °C
	Cold trap packing	Tenax-TA
	Cold trap holding time	5 min
	Trap heat temperature	280 °C
	Trap cool temperature	−20 °C
	Valve temperature	250 °C
	Interface heat temperature	250 °C
	In split	No
GC/MS	GC column	VB-1 (60 m × 0.25 mm × 1.0 μm)
	Initial temperature	40 °C (6 min)
	Oven ramp rate 1	4 °C/min (40–180 °C)
	Oven ramp rate 2	20 °C/min (180–250 °C)
	Final temperature	250 °C (10 min)
	Column flow	1.5 mL/min
	MS source temperature	200 °C
	Detector type	EI (Quadrupole)
	Mass range	Amu

**Table 2 ijerph-18-05508-t002:** Meteorological parameters and volatile organic compound (VOC) concentrations (μg/m^3^) in indoor and outdoor air.

	Indoor Air		Outdoor Air		I/O Ratio ^a^	R ^b^
	N	GM (GSD)	N	GM (GSD)		
**(a) Meteorological parameter**						
Temperature (°C)	30	26.0 (1.07)	13	25.5 (1.32)	1.03	0.34
RH (%)	30	60.8 (1.17)	13	64.0 (1.20)	0.93	−0.27
**(b) Aromatics**						
Benzene	30	2.34 (1.61)	13	2.03 (1.42)	1.14	0.53 **
Toluene	30	52.3 (2.58)	13	9.42 (3.10)	5.69	0.40 *
Ethylbenzene	30	7.03 (2.04)	13	2.16 (2.36)	3.09	0.35
*m,p,o*-Xylenes	30	9.64 (2.54)	13	2.69 (2.04)	3.48	0.69 **
Styrene	29	2.10 (2.54)	7	0.92 (1.99)	2.96	0.26
**(c) Aliphatics**						
*n*-Heptane	29	2.34 (2.55)	9	1.30 (1.56)	2.11	−0.19
*n*-Octane	30	2.38 (2.24)	10	0.89 (1.49)	3.25	−0.08
*n*-Nonane	30	1.72 (2.19)	12	0.67 (1.51)	2.55	−0.10
*n*-Decane	30	2.57 (2.14)	11	0.76 (1.50)	3.81	0.15
*n*-Undecane	29	1.87 (2.48)	6	0.56 (1.17)	5.18	−0.20
*n*-Dodecane	30	2.91 (2.67)	8	0.53 (1.04)	7.65	0.41
*n*-Tridecane	30	2.65 (2.38)	9	0.57 (1.04)	6.25	0.34
*n*-Tetradecane	30	4.31 (2.33)	10	0.56 (1.27)	9.21	−0.01
*n*-Pentadecane	29	1.18 (2.07)	7	0.75 (1.18)	1.98	−0.22
**(d) Terpenes**						
*α*-Pinene	30	2.01 (3.41)	4	0.70 (1.27)	6.74	−0.90 **
d-Limonene	29	4.15 (2.84)		ND		
**(e) Oxygenated VOCs**						
Nonanal	29	9.35 (2.11)	10	2.18 (1.32)	4.93	−0.01
*n*-Butanol	30	4.17 (2.22)	7	0.92 (2.08)	6.12	0.15
Formaldehyde	30	43.0 (2.11)	13	7.72 (1.74)	6.68	0.49 **
Acetaldehyde	30	8.01 (1.88)	13	2.49 (1.61)	3.23	0.54 **
Acetone	30	35.9 (2.74)	13	5.05 (2.31)	7.17	0.50 **
Benzaldehyde	30	1.21 (4.80)	12	0.45 (2.58)	4.22	0.42 *
Butyraldehyde	30	8.49 (3.37)	13	1.76 (3.51)	4.81	0.32
Propionaldehyde	29	1.44 (1.58)	8	0.94 (1.43)	1.64	0.77 **
24 VOCs total	30	257 (1.87)	13	43.6 (2.14)	5.88	0.53 **
TVOC	30	321 (2.15)	13	40.9 (2.74)	7.48	0.46 *

^a^ Indoor/outdoor ratio. VOCs are arranged by class, along with meteorological parameters, in descending I/O ratio order. ^b^ R, correlation coefficient of indoor and outdoor air (* *p* < 0.05, ** *p* < 0.01); N, number of samples; GM (GSD), geometric mean (geometric standard deviation) for the entire period and sites; temperature (°C); RH, relative humidity (%); ND, not detected; TVOC, total volatile organic compounds.

**Table 3 ijerph-18-05508-t003:** Factor loadings from the principal component analysis with varimax rotation.

	Factor 1	Factor 2	Factor 3	Factor 4	Factor 5	Factor 6
Acetone	**0.841**	0.155	0.078	0.092	−0.064	0.056
Butyraldehyde	**0.831**	0.110	0.113	0.047	−0.131	0.146
n-Pentadecane	**0.824**	0.213	0.203	0.326	0.195	0.105
Benzaldehyde	**0.797**	−0.026	0.139	0.274	0.350	0.001
Toluene	**0.744**	−0.057	0.305	0.072	0.016	0.304
d-Limonene	**0.693**	0.072	0.099	0.540	0.266	0.261
n*-*Butanol	**0.679**	0.058	0.137	0.451	0.438	0.119
Formaldehyde ^a^	**0.599**	0.002	0.201	0.367	0.439	−0.343
Styrene	**0.591**	−0.039	0.023	0.265	0.262	0.175
Acetaldehyde ^a^	**0.586**	0.284	−0.118	−0.010	0.472	0.388
n-Tetradecane	0.069	**0.978**	−0.073	−0.010	0.020	−0.031
Benzene ^a^	0.082	**0.883**	−0.062	0.200	−0.184	0.050
n-Undecane	−0.054	**0.791**	0.217	0.166	0.346	0.234
n-Dodecane	0.267	**0.707**	0.329	0.060	0.487	−0.043
n-Tridecane	0.169	**0.654**	0.392	−0.084	0.523	−0.098
n-Nonane	0.017	0.095	**0.889**	0.098	0.360	0.088
n-Heptane	0.215	0.035	**0.882**	0.204	−0.062	0.114
n-Octane	0.543	0.108	**0.755**	0.130	0.204	0.033
α-Pinene	0.235	0.084	0.175	**0.816**	−0.050	0.172
Nonanal	0.615	0.163	0.286	**0.631**	0.234	−0.013
Propionaldehyde	0.402	0.255	0.101	**0.606**	0.338	0.169
n-Decane	0.143	0.176	0.333	0.217	**0.802**	0.276
Ethylbenzene	0.308	0.018	0.150	0.198	0.145	**0.848**
*m,p,o*-Xylenes	0.595	0.138	0.194	0.359	0.113	**0.598**
Eigenvalues	6.78	3.64	3.00	2.69	2.62	1.81
% variance	28.2	15.2	12.5	11.2	10.9	7.55
Cumulative % variance	28.2	43.4	55.9	67.1	78.0	85.6
Possible source(s)	Emissions within the store	Indoor fuel use, floor and wall coverings	Outdoor fuel use, floor and wall coverings	Fossil fuel combustion and cleaning products	Solvent use and surface coatings	Vehicle emissions, painting/ printing products

^a^ High carcinogenic risk; used to assess the excess cancer risk (ECR) in Table 6. Boldface denotes high factor loadings considered to be marker species.

**Table 4 ijerph-18-05508-t004:** Correlations between factors and outdoor air ^a^ and mean indoor to outdoor (I/O) ratios.

	Factor 1	Factor 2	Factor 3	Factor 4	Factor 5	Factor 6
Factor 1	1					
Factor 2	0.274	1				
Factor 3	0.500 **	0.311	1			
Factor 4	0.703 **	0.335	0.544 **	1		
Factor 5	0.387 *	0.475 **	0.539 **	0.516 **	1	
Factor 6	0.670 **	0.263	0.451 *	0.648 **	0.481 **	1
Outdoor 24 VOCs	0.521	0.012	0.184	0.412 *	0.367 *	0.614 **
I/O ratio ^b^	6.22	4.13	1.86	3.93	3.81	3.37

^a^ Used the sum of the marker species concentrations for each factor and the total of all 24 species concentrations for the outdoor air. *p*-value: ** *p* < 0.01, * *p* < 0.05. ^b^ Used the total of all 24 species for both indoor and outdoor concentrations.

**Table 5 ijerph-18-05508-t005:** Contributions of volatile organic compound (VOC) sources ^a^ by underground shopping district (USD) environmental factors.

	Number of Data Points	24 VOCs Total			VOC Concentration (μg/m^3^) ^b^						Contribution (%) ^c^				
		Indoor (μg/m^3^)	Outdoor (μg/m^3^)	I/O ratio	Total	Store	Indoor Fuel	Outdoor Fuel	Comb. C-Prod.	Vehicular	Store	Indoor fuel	Outdoor fuel	Comb. C-Prod.	Vehicular
**(a) Overall**	30	257	43.6	5.88	237	184	14.8	6.49	12.0	17.0	77.7	6.26	2.74	5.07	7.17
**(b) Location**															
Gyeonggi	5	550	89.5	6.15	547	457	20.6	12.4	20.2	32.8	83.6	3.78	2.26	3.69	6.00
Daejeon	2	225	49.3	4.57	224	195	8.13	4.28	8.31	6.92	86.9	3.63	1.91	3.71	3.09
Daegu	6	235	47.9	4.90	223	176	12.1	6.19	8.97	16.2	78.8	5.41	2.78	4.03	7.29
Busan	10	202	33.5	6.04	185	141	13.7	5.38	12.9	10.0	76.2	7.42	2.90	6.99	5.40
Gwangju	2	180	65.5	2.75	174	126	8.08	3.67	11.9	23.6	72.4	4.64	2.10	6.85	13.5
Suncheon	2	103	8.87	11.7	101	41.9	22.8	2.65	1.09	31.0	41.5	22.6	2.63	1.08	30.7
Jeju	3	487	54.9	8.87	482	353	27.7	16.0	44.1	35.2	73.2	5.74	3.32	9.15	7.29
**(c) Type of USDs**															
Open	2	180	65.5	2.75	174	126	8.08	3.67	11.9	23.6	72.4	4.64	2.10	6.85	13.5
Semi-open	28	263	42.4	6.20	242	189	15.5	6.76	12.0	16.6	77.9	6.38	2.79	4.95	6.84
**(d) Type of stores**															
Open	14	319	47.6	6.70	311	257	11.9	8.03	15.9	16.2	82.5	3.83	2.58	5.11	5.21
Mix	9	313	60.3	5.19	300	222	17.8.	9.83	17.6	28.6	74.0	5.94	3.28	5.89	9.55
Closed	7	128	24.2	5.30	110	74.2	18.1	2.49	4.16	9.50	67.2	16.5	2.25	3.77	8.61
**(e) Number of stores**															
<99	10	192	34.7	5.53	171	130	9.42	4.10	7.60	17.6	76.1	5.49	2.39	4.43	10.3
100–199	10	275	41.8	6.56	255	202	18.8	6.20	13.4	11.9	79.4	7.39	2.44	5.25	4.66
≥200	10	321	57.4	5.60	308	235	18.3	10.7	16.9	23.4	76.4	5.95	3.49	5.50	7.59

^a^ Store, emissions within the store; Indoor fuel, indoor fuel use, floor and wall coverings; Outdoor fuel, outdoor fuel use, floor and wall coverings; Comb. C-Prod., fossil fuel combustion and cleaning products; Vehicular, vehicular emissions, painting/printing products. ^b^ Sum of the marker species concentrations for each source; Total, 24 VOCs total. ^c^ Concentration of each source divided by the total VOC concentration (24 VOCs).

**Table 6 ijerph-18-05508-t006:** Excess cancer risk (ECR) for the 95th percentile concentration of benzene, formaldehyde, and acetaldehyde.

	IRIS Group	Toxicity Value (μg/m^3^)^−1^	Indoor Air		Outdoor Air	
			Conc.	ECR	Conc.	ECR
Benzene	A	2.2 × 10^−6^ − 7.8 × 10^−6^	4.46	8.6 × 10^−7^ − 3.1 × 10^−6^	3.61	7.0 × 10^−7^ − 2.5 × 10^−6^
Formaldehyde	B1	1.3 × 10^−5^	104	1.2 × 10^−4^	14.6	1.7 × 10^−5^
Acetaldehyde	B2	2.2 × 10^−6^	24.0	4.6 × 10^−6^	4.13	8.0 × 10^−7^

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
