# Peer review of "Volatile Organic Compounds in Underground Shopping Districts in Korea"

_ijerph, 2021, doi:10.3390/ijerph18115508_

Round 1

Reviewer 1 Report

Dear Authors,

I found the reviewed manuscript undoubtedly more clear and effective.

Cross fingers,

Author Response

Thank you for your comments.

Reviewer 2 Report

This is an interesting study focusing on the air pollution in underground shopping districts, where usually have poor ventilation, probably dangerous to the salesman. This version has well revised the existed issues. Minor revisions are needed before accepted for publication.

  1. Although the authors have reported that they sampled the out door air pollutants near the USDs, the specific information of location for sampling has not provided. I suggest the author to provide this information, such as the distance to the USDs, hight, etc.
  2. Due to the different constructed year of these USDs, the authors also need to controll this when discusss the difference of indoor air pollutants particularly when assessing the VOCs.
  3. I suggest the authors the correlate this indoor air pollutants with the health of salesman in future studies.

Author Response

This is an interesting study focusing on the air pollution in underground shopping districts, where usually have poor ventilation, probably dangerous to the salesman. This version has well revised the existed issues. Minor revisions are needed before accepted for publication.

Response: We thank the reviewer for positively evaluating our efforts for revising the manuscript, and providing valuable comments again. Because the format of the response is the same as the previous one, you can refer to the explanation of the previous response if you’d like to conform it. For convenience, the comments are italicized and numbered consecutively. The line (L) numbers in the responses correspond to those in the revised manuscript. The changes are underlined. The “track changes” function was used to revise the manuscript so that all the changes are easily visible in the revised manuscript. The important changes are additionally highlighted (in yellow) and red text.

  1. Although the authors have reported that they sampled the outdoor air pollutants near the USDs, the specific information of location for sampling has not provided. I suggest the author to provide this information, such as the distance to the USDs, height, etc.

We have clearly rewritten the outdoor air sampling information, as follows: “We sampled outdoor air at one point near within 10 m distance the entrance of each USD.” (L113-114).

  1. Due to the different constructed year of these USDs, the authors also need to control this when discusses the difference of indoor air pollutants particularly when assessing the VOCs.

In the opinion of the reviewer, it is correct that the indoor VOCs is related to the year of construction. However, most of the USDs we measured are over 20 years old except for one. Therefore, we were not considered the correlation between the indoor VOCs and the construction year because we did not take into the VOCs emitted from the building materials.

  1. I suggest the authors the correlate this indoor air pollutants with the health of salesman in future studies.

In the future, we plan to conduct a risk assessment through a questionnaire on the health impacts of USDs workers. We noted the following in the conclusion: “In addition, it is necessary to assess practical risk by setting exposure scenarios through the health impact questionnaire of workers in USDs.” (L465-466).

Reviewer 3 Report

1. Lines 217-226: It is still confusing to understand the differences between these two classifications for USD.

2. Lines 478-480: It is inadequate to mention that the sum of the total peak area eluted from n-hexane to n-hexadecane is TVOCs because not all components of VOCs are collected by the method “ISO16000-6”. For instance, the TVOCs does not include other components of nonanal, n-butanol, α-pinene, and d-limonene that required other method to measure. It is better to report all compounds consisted of the TVOCs in this study.

3. Lines 540-544: It is insufficient to estimate carcinogenic risk assessment only relying on the multiplying the inhalation value (i.e., slope factor or unit risk) with the 95th percentile concentrations of the hazardous materials. In the final step of risk assessment (risk characterization), the assumption and scenarios have to be mentioned (such as exposure frequency, exposure duration and expected lifetime) that can be used for comparison with other findings. Different scenarios will produce different results.

4. The table numbers in this revision still are not corrected.

Reviewer 4 Report

Please, see attached file.

Round 2

Reviewer 3 Report

  1. Lines 1069-1073: It must describe the assumption and scenarios to estimate excess cancer risk in this study. As mention in the EPA IRIS, the inhalable cancer risk is based on a standard air intake of 20 m3 per day, a standard body weight of 70 kg for an adult human, and 50% absorption via inhalation over 70-year exposure duration.

Author Response

This manuscript is a resubmission of an earlier submission. The following is a list of the peer review reports and author responses from that submission.

Round 1

Reviewer 1 Report

-

Author Response

Dear reviewer, 

We thank the reviewer for providing a good summary of our study and valuable comments. 

Please see the attachment for the following are our responses to the comments.

Best regards.

Reviewer 2 Report

Introduction

  1. Lines 46-48: Please clarify the carcinogen class specifically for formaldehyde, benzene, and acetaldehyde using the USEPA system.
  2. Lines 63-65: The sentence is unclear and must be rewritten.

Methods

  1. Lines 77-83: How to separate the “open” type in both two classification? It is better to provide figures for explanation.
  2. Lines 90-94: Please provide the detailed information about the range of store areas and stores using air conditioners.
  3. Lines 100-104: Why were these 24 VOCs selected for analyses?
  4. Lines 125-127: “TVOC concentrations were calculated by applying the retention factor (RF) of toluene to the sum of the total peak area eluted from n-hexane to n-hexadecane.” Was the TVOCs calculated without considering other 6 components using different analyzing method? How about other components, such as nonanal, n-butanol, α-pinene, and d-limonene? The definition of TVOC is required to be clarified. Additionally, please provide the citation for this approach.
  5. Lines 165-167: It is too easy to be understood for carcinogenic risk assessment. The assumption and scenarios have to be mentioned before applying to estimate the risk, such as exposure frequency, exposure duration and expected lifetime for different subpopulation.

Results and Discussion

  1. Table 2: Please clarify why the indoor and outdoor concentrations of 24 VOCs were lower than the TVOC, which is eluted only from n-hexane to n-hexadecane.
  2. Line 233: The table number should be corrected as Table 3. Why only 6 VOCs components were described in selected stores?
  3. Line 349: The table number has to be corrected as Table 4.
  4. Line 352: Please correct the table number as Table 5.
  5. Line 356: The table number should be corrected as Table 6. Additionally, the “Total” indoor and outdoor results are confusing with those of “24 VOCs total” in Table 2. Please clarify the terms used in this study.
  6. Lines 364-366: The classification of formaldehyde and acetaldehyde must be clarified (i.e., B1 or B2) here.
  7. Line 386: Please correct the table number as Table 7.
  8. Lines 389-391: The sentence is not clear and has to be rewritten.

Author Response

(The authors gave the same response as above.)

Reviewer 3 Report

It’s an interesting study concerning the air pollution in underground shops. But there are few concerns belowing.

  1. In this study, the measurements were conducted from July to October. I wonder are there any difference of concentration under different indoor temperatures. Higher temperature may promote the releasing of VOCs. Therefore, the authors may also need to consider the effect of temperature and seasons.
  2. In this study , the authors mainly measured the VOCs concentration in underground shops. It would be better to compare this with the VOCs concentration in shops on the ground.
  3. Usually, the indoor measuring number of locations need to be decided by the square meter of the area. More locations will be needed if the area is too big.

Author Response

(The authors gave the same response as above.)

Reviewer 4 Report

Dear authors,

I read carefully your interesting manuscripts. Here below some observations on which you should spend some efforts

  • You should specify, for each investigated store, the number of measurement points;
  • You are invited to compare Korean limit values with other limit values in the world (e.g. ASHRAE, ACGIH, and so on);
  • Sentences at P14-L413->L415 are unclear;
  • You are invited to refine the English.

Best regards

Author Response

(The authors gave the same response as above.)
